# Soft Selective Sweep on Chemosensory Genes Correlates with Ancestral Preference for Toxic Noni in a Specialist *Drosophila* Population

**DOI:** 10.3390/genes12010032

**Published:** 2020-12-29

**Authors:** Erina A. Ferreira, Sophia Lambert, Thibault Verrier, Frédéric Marion-Poll, Amir Yassin

**Affiliations:** 1Laboratoire Évolution, Génomes, Comportement et Écologie, CNRS, IRD, Université Paris-Saclay, 91198 Gif-sur-Yvette, France; ferreira@egce.cnrs-gif.fr (E.A.F.); frederic.marion-poll@egce.cnrs-gif.fr (F.M.-P.); 2Institut Systématique Evolution Biodiversité (ISYEB) Centre National de la Recherche Scientifique, MNHN, Sorbonne Université, EPHE 57 rue Cuvier, CP 50, 75005 Paris, France; slambert@bio.ens.psl.eu (S.L.); thibault.verrier@free.fr (T.V.); 3AgroParisTech, Université Paris-Saclay, 75231 Paris, France

**Keywords:** insect-plant interactions, standing genetic variation, genome-wide selection scan, gene family evolution, feeding behavior

## Abstract

Understanding how organisms adapt to environmental changes is a major question in evolution and ecology. In particular, the role of ancestral variation in rapid adaptation remains unclear because its trace on genetic variation, known as soft selective sweep, is often hardly recognizable from genome-wide selection scans. Here, we investigate the evolution of chemosensory genes in *Drosophila yakuba mayottensis*, a specialist subspecies on toxic noni (*Morinda citrifolia*) fruits on the island of Mayotte. We combine population genomics analyses and behavioral assays to evaluate the level of divergence in chemosensory genes and perception of noni chemicals between specialist and generalist subspecies of *D. yakuba*. We identify a signal of soft selective sweep on a handful of genes, with the most diverging ones involving a cluster of gustatory receptors expressed in bitter-sensing neurons. Our results highlight the potential role of ancestral genetic variation in promoting host plant specialization in herbivorous insects and identify a number of candidate genes underlying behavioral adaptation.

## 1. Introduction

Host plant specialization by herbivorous insects is a complex phenomenon requiring the simultaneous evolution of multiple adaptive phenotypes on the same genome. Traditionally, these phenotypes are classified under two broad categories: preference phenotypes inducing the choice of the particular host by the insect, and performance phenotypes improving the survival of the insect on the host [1]. Preference phenotypes could rely on visual, chemical, anatomical or phenological attributes of the host plant. The signals of each of these attributes need to be transmitted by the peripheral nervous system of the insect to the central nervous system, which following processing of the information would elicit attraction or repulsion behaviors. How the insect perceives the attributes of its host plant is a question of intense evolutionary and neurogenetic research [2,3,4].

Much of our knowledge on the genetic basis of perception of environmental cues comes from studies on the model fly *Drosophila melanogaster*. The family Drosophilidae contains a wide spectrum of fly-plant associations going from generalist detritivorous species such as *D. melanogaster* to strict herbivorous such as species of the genus *Scaptomyza* [5]. Comparative genomics of the chemosensory gene families between generalist and specialist taxa have provided significant insights on how host plant shift and specialization can be driven by these genes [6,7,8]. There are multiple gene families that encode transmembrane receptors localized on the peripheral nervous system, of which three have attracted much attention due to their high diversification rate, namely olfactory (ORs), gustatory (GRs) and ionotropic (IRs) receptors. A fourth family, odorant-binding-proteins (OBPs), encodes for proteins that bind with both volatile odors and soluble tastants to help them migrate within the intercellular hemolymph up to the chemosensory neuron.

Between-species comparative genomics revealed the dynamic evolution of each of chemosensory gene families in terms of gene gains through duplication and conversion, gene losses through deletions and pseudogenizations, and protein sequence evolution through extensive non-synonymous mutations [9,10]. Specialist species are often characterized by a higher rate of gene losses and protein changes. At the intra-specific level, which should correspond to the early stages of host shift, only very few cases have been identified, e.g., the many geographical races of the cactophilic species *Drosophila mojavensis* in North America [11,12]. Of particular interest is the parallel specialization of the species *Drosophila sechellia* and the subspecies *Drosophila yakuba mayottensis* on toxic noni (*Morinda citrifolia*) fruits on separate islands of the Indian Ocean, i.e., the Seychelles and Mayotte, respectively [13,14]. Noni toxicity is due to its high content of medium-chained carboxylic acids (hexanoic and octanoic acids) in its fruit [15]. Remarkably, *D. sechellia* has evolved a strong olfactory attraction to those acids [16,17], as well as to methyl hexanoate, the major ester characteristic of the rotten, non-toxic fruit of noni [18]. Genetic studies revealed that evolutionary changes at the ionotropic receptor *Ir75b* and at the olfactory receptor *Or22a* underlie *D. sechellia* preference to noni hexanoic acid and methyl hexanoate, respectively [19,20]. Matsuo et al. [21] also suggested that the odorant-binding-protein OBP57e/d may also play a role in the gustatory preference of this species to noni acids. However, in spite of evidence for rapid evolution of gustatory receptors in *D. sechellia* [6,7], the role of those receptors in noni specialization remains unknown. This may be partly because both sugar and bitter sensing neurons may be involved in dose-dependent acid sensing, as it has recently been demonstrated from functional studies in *D. melanogaster* [22], therefore complicating the dissection of this character. For *D. y. mayottensis*, an olfactory preference for noni fruits in adult flies exists [14], but we still do not understand of the chemosensory evolution underlying this preference, as well as possible gustatory preference, in this subspecies.

Intra-specific adaptive changes related to host shift could be detected through genome-wide selection scans (GWSS) comparing two populations or races with different hosts [2,4]. However, our ability to detect such changes through GWSS depends on multiple factors, most importantly, the frequency of the selected allele(s) in the ancestral population before encountering the new host. If the advantageous allele was present at a very low frequency in the ancestral population or was introduced by a new mutation, selection driving it to fixation or near fixation on the new host will leave a strong signal on the genome known as a ‘hard selective sweep’. Depending on the rate of recombination, neutral alleles linked to the selected one will also increase in frequency facilitating the detection of the whole region through GWSS using large genomic windows. However, if the advantageous allele was already at intermediate frequency in the ancestral population, its detection through GWSS becomes more complicated, because fewer neutrally-linked loci will be associated to its fixation, a phenomenon known as ‘soft selective sweep’ [23,24].

In a previous GWSS study, we have used differentiation at 10-kb windows in highly-recombining regions to detect ‘hard selective sweeps’ associated with specialization on toxic noni in *D. yakuba* [14]. Although, we have identified multiple regions, surprisingly, none contained any member of the four chemosensory gene families, despite a significant difference in olfactory preference between generalist and specialist populations. In this paper, we test the hypothesis that chemosensory genes in *D. y. mayottensis* might be under ‘soft selective sweeps’ and that an ancestral, yet substantial, preference for noni chemicals may be present in populations from the ancestral range. Using a combination of population genomics and behavioral analyses, we confirmed both hypotheses and identified some candidate genes potentially involved on the specialization on noni in both *D. yakuba* and *D. sechellia*. We then discuss the relevance of our results within the broader context of plant-insect interaction and adaptation.

## 2. Materials and Methods

### 2.1. Population Genomics Analyses of Chemosensory Genes

We used genomic data from Yassin et al. [14] produced from two pools of 22 isofemale lines of *D. y. mayottensis*. Each pool consisted of 33 F_1_ females from 11 lines (i.e., 3 females per line). As in Yassin et al. [14], we used sequences produced by Rogers et al. [25] for 10 inbred lines of *D. y. yakuba* from Kenya and Cameroon. All reads were mapped to the *D. yakuba* reference genome v.1.05 obtained from Flybase (https://flybase.org/; [26]) using Minimap2 software package [27]. Minimap2-generated SAM files were converted to BAM format using samtools 1.9 software [28]. The BAM files were then cleaned and sorted using Picard v.2.0.1 (http://broadinstitute.github.io/picard/). However, instead of directly merging both *D. y. mayottensis* pools as in Yassin et al. [14], we first generated using Popoolation 2 [29] a synchronized mpileup file for the two pools. We then used a customized Perl script to extrapolate allele frequencies to 22 diploid counts for each pool, and by excluding sites with less than 3 reads. The two genotyped pools were then merged for subsequent analyses. We also generated synchronized files for the 20 *D. y. yakuba* lines using Popoolation 2. However, to account for residual heterozygosity in those inbred lines which was not considered in Yassin et al. [14], we genotyped each line to obtain its diploid alleles, after excluding sites with less than 3 reads and alleles with frequencies less than 25% for the total counts using a customized Perl script. We also excluded tri-allelic sites for those inbred lines. For each geographical population, the diploid genotypes were then pooled and a synchronized file using the two *D. y. yakuba* populations and the *D. y. mayottensis* population was then generated. Sites with less than 10 counts at any of the three populations were excluded. This file was used to estimate nucleotide diversity (*π*) and pair-wise *F_ST_* estimates at each site using HSM [30] formula introduced in a Perl script. We obtained a list of chemosensory genes, their coordinates and their orthology to the *D. melanogaster* genome from FlyBase. For genes with multiple orthologs due to paralogy, we used BLAST software [31] to choose the ortholog with the highest hit. For each gene, we averaged *π* and *F_ST_*, after including up- and downstream 2-kb regions to account for possible regulatory sequences relevant to the gene function. Perl scripts are provided in Appendix A.

In order to estimate deviation of each gene from neutral expectations, we first inferred a demographic model from presumably-neutral short autosomal short introns between *D. y. mayottensis* and the Kenyan population of *D. y. yakuba* using the “prior_onegrow_mig” model implemented in the DADI ver. 1.7. software package [32] as in Yassin et al. [14]. Based on the most optimal model parameters and theta estimates inferred from DADI, we conducted 10,000 simulations of a 5 kb region (i.e., amounting to the average length of a chemosensory gene ± 2 kb) using the *msms* software package (https://www.mabs.at/ewing/msms/index.shtml, [33]. After considering a recombination rate of 2.5 × 10^−8^ corresponding to the average rate in *D. melanogaster* [34], the *msms* command was: ms 54 10,000 -t 67 -r 341.9534019 5001 -I 2 10 44 0 -n 1 1.345 -n 2 0.223 -eg 0 2 34.23591156 -ma x 0.205 0.205 x -ej 0.053 2 1 -en 0.43275868 1 1.

For each run, we estimated *π* in the two populations and *F_ST_* between them, both averaged over all segregating loci using a Perl script. We also estimated the highest *F_ST_* value at a site for each run (hereafter *F_ST_max_*). The distribution of *F_ST_max_* was then plotted to identify the 0.95 quantile, i.e., the value above which the hypothesis of a neutral differentiation at a given site could be rejected at *p* < 0.05, as in Bastide et al. [35]. We also reanalyzed the simulation outcome to count the number of sites, exceeding the neutral *F_ST_max_* threshold.

### 2.2. Behavioral Feeding Preference Analyses

Our previous behavioral analysis in *D. yakuba* [14] involved testing long-range olfactory preference through releasing and recapturing flies using a choice between traps including banana or noni fruits. However, because the most differentiating chemosensory genes were found to be those encoding gustatory receptors (see below), we decided to evaluate the gustatory preferences using multiple capillary feeders (MultiCAFE) technique [36]. We tested the same two *D. yakuba* populations used in Yassin et al. [14], i.e., a strain of *D. y. mayottensis* collected from the Bay of Soulou in Mayotte in 2013 and a strain of *D. y. yakuba* collected from Kunden in Cameroon in 1967. We also used a strain of *D. sechellia* collected from the Seychelles in 1985 as in Yassin et al. [14]. In addition, we used a strain of *D. melanogaster* collected from Mt. Oku in Cameroon in 2016 (courtesy of Jean R. David), as well as a laboratory strain iso-1, which carried mutations in some of the genes of interest identified in this study (courtesy of Jean-Luc Da Lage). All these strains were kept in a constant size (N = ~2000 flies) on a standard *Drosophila* medium at 18 °C.

For the MultiCAFE test, 10 > 5-days-old females were sorted the day before the experiment and placed for starvation in tubes with a filter paper humidified with 2 mL of distilled water. The tubes were kept for 24 h in a dark incubator at 25 °C. Flies were then aspirated and placed in group in custom feeding chambers (5.5 × 0.6 × 0.5 cm) and six 5 µL capillaries were introduced through small holes in the ceiling of the chamber, alternating between those containing the control solutions with 30 g·L^−1^ sucrose and test solutions containing 30 g·L^−1^ sucrose mixed with methyl hexanoate (MH; Sigma, St Quentin Fallavier, France; CAS number 106-70-7), hexanoic acid (HA; Sigma; CAS number 142-62-1) or octanoic acid (OA; Sigma; CAS number 124-07-2). HA and OA are the main noni toxins that are abundant at the toxic ripe stage, whereas MH is the characteristic ester of the non-toxic overripe stage [15,18]. All substances were tested at 0.5% concentration (4.6 g·L^−1^), and four replicates per strain were tested (except for *D. melanogaster* iso-1 mutant strain which was not tested for this concentration). HA was also tested at 1% concentration (9.3 g·L^−1^) due to its higher solubility and lesser toxicity compared to OA. For this analysis, capillaries were introduced as pairs into individual feeding chambers (2.3 × 0.6 × 0.5 cm), and multiple individuals per strain were quantified: 27, 33, 70, 74 and 30 for *D. y. mayottensis*, *D. y. yakuba*, *D. sechellia*, *D. melanogaster* Oku and *D. melanogaster* iso-1, respectively. All solutions were colored with Brilliant Blue R (Sigma; CAS Number 6104-59-2) to facilitate consumption quantification.

Flies were videotaped in MultiCAFE chambers at 25 °C for 2 h, using Logitech HC920 webcams, by capturing one image per minute using an open source camera security software, Ispy (https://www.ispyconnect.com/). The stacks of images were analyzed using a custom plugin developed in Java to work with Icy [37] to calculate consumption of each solution. The evaporation was estimated by measuring changes in the liquid level of 2–4 capillaries placed in the same cages, but out of reach for the flies. At the end of the experiment (i.e., 2-h), a preference index (PI) was calculated for each substance as follows:(1)$$PI\;=\;\frac{\left(C_{S}\;-\;C_{C}\right)}{\left(C_{S}\;+\;C_{C}\right)}$$
where *C_S_* and *C_C_* are consumption from the substance and control capillaries, respectively, and computed as the actual volume drop minus drop observed in the evaporation capillaries in µL. For flies tested in groups, *C_S_* and *C_C_* were respectively averaged per experiment. Statistical analyses were conducted using the R package [38]. For analyses at 0.5% concentrations, we used pairwise Wilcoxon test, since only four values per strain per substance, i.e., the average consumption by a group of 10 flies, were compared. For analyses at 1%, since consumptions were measured individually, we used Student’s *t* test for pairwise comparisons, in addition to the Wilcoxon test.

## 3. Results

### 3.1. A Handful of Chemosensory Genes, Mostly Encoding Gustatory Receptors, Deviate from Neutral Expectations

On average, the level of genetic variation for chemosensory genes differed between Mayotte and Kenya *D. yakuba* populations (*π* = 0.0095 and 0.0107, respectively; Kruskal-Wallis test *p* < 0.01) but not between gene families in Mayotte (Kruskal-Wallis test *p* = 0.51). Mean genetic differentiation between the two populations also did not significantly differ among the four chemosensory gene families (averaged *F_ST_* = 0.08; Kruskal-Wallis test *p* = 0.46; Figure 1a). Remarkably, these values were far lower than the average neutral *F_ST_* expected from the msms simulations (*F_ST_* = 0.136; Figure 1b). Interestingly, *F_ST_* for a single gene exceeded the 95% quantile estimate (F_ST_ = 0.161) from the *msms* simulation. This gene was the gustatory receptor *Gr22c* (*F_ST_* = 0.168). It was followed by its two adjacent paralogs *Gr22b* (*F_ST_* = 0.143) and *Gr22d* (*F_ST_* = 0.147). The three genes belong to a cluster of closely-related GR paralogs (hereafter the Gr22a clade) falling on the distal part of chromosome arm 2L. This clade contains six genes that are expressed in bitter tasting organs [39]. *Gr22c* was pseudogenized in noni-specialist *D. sechellia* and lost in Pandanus-specialist *D. erecta* [6]. *Gr22b* showed a particularly rapid non-synonymous substitution in *D. sechellia* (*ω* > 1, [7]). Moreover, two other genes *Gr22d* and *Gr22f* have also been pseudogenized in *D. sechellia* [6]. For the other families, a single exception was noticed for the odorant-binding-protein gene *Obp46a* (*F_ST_* = 0.151).

*msms* simulations based on the demographic model inferred from presumably neutral short autosomal intronic sites indicated that the probability to obtain a SNP with *F_ST_max_* ≥ 0.90 in the absence of selection was *p* ≤ 0.05 (Figure 1c). We found 13 sites crossing this threshold. Of which, 2, 2 and 9 belonged to GR, IR, and OR families, respectively (Table 1). Those 13 sites fell within 5 genes, with some genes having more than one presumably adaptively evolving site. The probability of having more than one site crossing the 0.90 threshold for one gene is strongly skewed (Figure 1d). Indeed, three genes had only a single site with *F_ST_* ≥ 0.90, i.e., *Gr22b*, *Gr93c* and *Or13a*. One gene, *Ir7a*, had two deviating sites (*p* < 9 × 10^−3^). However, the most interesting finding was the olfactory receptor *Or22a*, which underlies *D. sechellia* response to noni major ester ([20], where 8 sites deviated from neutral expectation. According to the *msms* simulations, the probability to obtain eight sites with *F_ST_* ≥ 0.90 in a single gene under neutral conditions is very small (*p* < 3 × 10^−4^). When we considered a less stringent threshold of *F_ST_* ≥ 0.86, which corresponded to the 0.90 quantile of the *msms* simulations, three additional sites, all falling in gustatory receptor genes, namely *Gr22d*, *Gr59a* and *Gr93b*, were found (Table 1). No site at OBP genes was found.

Our *F_ST_* values correspond to differentiation between a pair of populations, but they cannot tell, on their own, which of the two populations has experienced most differentiation since splitting from the common ancestor. Consequently, we checked the frequency of differentiating alleles between Kenyan *D. y. yakuba* and Mayotte *D. y. mayottensis* in another *D. y. yakuba* population from the presumably the ancestral region of the species, i.e., Cameroon [40]. We found that only sites at *Gr22b*, *Gr22d*, *Gr59a* and *Gr93b* had a differentiation between *D. y. mayottensis* and *D. y. yakuba* from Cameroon ≥ 0.50, i.e., the differentiating alleles at those genes have specifically increased in frequency only in *D. y. mayottensis*. For *Or22a* and the other genes, all *D. y. mayottensis* fixed alleles were more frequent in Cameroon than Kenyan alleles (Figure 2a,b). This suggests that those alleles did not likely originate in Mayotte.

The overlap between genes of the Gr22a clade in our gene-level and population-specific site-level analyses, as well as the rapid evolution of this clade in *D. sechellia*, motivated us to investigate more thoroughly its polymorphism in *D. y. mayottensis* (Figure 2c). In *Gr22b*, the most differentiating site is a synonymous mutation that does not affect protein sequence. For *Gr22d*, the most differentiating site falls ~0.5 kb downstream the gene. Interestingly, it is close to a nonsense mutation in the second exon of *Gr22d* at 2L:1,764,799. This non-sense mutation has a high intermediate frequency in *D. y. mayottensis* (~0.70%) but does not reach fixation. In *D. y. yakuba*, the mutation is present in both Kenyan and Cameroonian populations but at very low (0.10%) to low (0.15%) frequencies, respectively (Figure 2d). The genetic differentiation pattern observed at *Gr22d* therefore could be explained by selection on the most-differentiating sites, if they play a role in the regulation of *Gr22d* or other genes of the Gr22a clade, on the nonsense mutation itself, or on the epistatic interactions between the most-differentiating sites and the nonsense mutation. Because our gene-level analyses also included 2kb up- and downstream sequences for each gene, the high average *F_ST_* values for *Gr22c* that we found above is mostly due to the high differentiation sites in *Gr22b* and *Gr22d*, since the transcribed sequence of this gene does not include any particularly differentiating sites.

### 3.2. Generalist D. yakuba Has a Substantial Dose-Dependent Gustatory Preference for Noni Toxins

Among the three noni chemicals tested at 0.5% concentration (4.6 g·L^−1^), no feeding preference was found for MH, the characteristic ester that increases in the rotting stage (Figure 3a). However, significant pairwise differences, especially between the generalist *D. melanogaster* and the specialist *D. sechellia*, were observed for the two toxins OA and HA (Wilcoxon’s test *p* = 0.029; Figure 3b,c), characteristic of the toxic ripe stage. No significant difference between *D. y. yakuba* and *D. y. mayottensis* was found for either toxin. However, *D. y. mayottensis* did not significantly differ from specialist *D. sechellia* for either toxin, whereas *D. y. yakuba* suggestively differed from D. sechellia in preference for OA (Wilcoxon’s test *p* = 0.057), the most lethal toxin, but not for HA (Figure 3b,c).

We also conducted an experiment using a higher concentration (9.3 g·L^−1^) of HA. Here, both *D. sechellia* and *D. y. mayottensis* showed a significant preference for the acid solution compared to their respective, generalist relatives, i.e., *D. melanogaster* (*t* = −2.708, d.f. = 141.42, *p* = 0.008; Wilcoxon’s test *p* < 0.010) and *D. y. yakuba* (*t* = −2.133, d.f. = 57.76, *p* = 0.037; Wilcoxon’s test *p* = 0.052), respectively, although the difference was more pronounced between *D. sechellia* and *D. melanogaster* (Figure 3d). Interestingly, whereas no significant difference was found between *D. sechellia* and *D. y. mayottensis* (*t* = −0.060, d.f. = 61.21, *p* = 0.952; Wilcoxon’s test *p* = 0.670), *D. sechellia*’s preference was significantly higher than that of *D. y. yakuba* (*t* = 2.251, d.f. = 62.43, *p* = 0.028; Wilcoxon’s test *p* = 0.019) We also compared wild-type *D. melanogaster* with the strain iso-1, which carries non-sense mutations at both *Gr22b* and *Gr22d*. Here, a strongly significant difference was observed, with the mutant strain showing almost no preference for the acidic solution (*t* = −7.9002, d.f. = 63.76, *p* = 4.9 × 10^−11^; Wilcoxon’s test *p* = 9.5 × 10^−10^; Figure 3d).

## 4. Discussion

### 4.1. Contrast between Olfactory and Gustatory Evolution in D. yakuba

Our results support the hypothesis that soft selective sweep on ancestral genetic variation in chemosensory genes might have promoted the evolution of *D. y. mayottensis* preference for noni chemicals, but they also unravel a major contrast in olfactory and gustatory evolution between molecular and phenotypic levels. On the one hand, specialist *D. yakuba* significantly differ in olfactory preference for noni from generalist flies [14] but there is weak signal of selection on olfactory receptors. On the other hand, gustatory receptors are significantly present in the pool of non-neutrally evolving SNPs although only suggestive, not significant, gustatory differences exist between specialist and generalist *D. yakuba*. The contrast at the phenotypic level may be explained by differences in the experimental settings we used for the two behaviors.

First, we measured olfactory preference by releasing flies from a distance from two traps each with a different fruits (i.e., banana vs. noni) [14]. Evolved repulsion against banana, for example, could have coupled with preference for noni to increase the difference between Mayotte and the mainland. No choice between two fruits was given to the flies in the gustatory experiment, and it could be that medium-chained carboxylic acids at the tested concentration were general appetizers. For example, even the generalist *D. melanogaster* had always a positive preference index for capillaries with noni chemicals (including the toxins) in agreement with previous reports in this species [22,41].

Second, flies were “trapped” by the choice they made in the olfactory experiments leaving no room for learning or plasticity (e.g., after consumption of a certain amount) [42]. This was not the case for the gustatory experiment where flies were confined in one chamber with capillaries containing alternative solutions placed only a few millimeters apart.

Third, the olfactory experiment used whole noni fruits with mixtures of chemicals, whereas only a single chemical was used per gustatory experiment. For oviposition site choice behavior, which is partly determined by gustatory perception, it has been noticed that mash of noni fruits, but not hexanoic or octanoic acids, elicits oviposition in *D. melanogaster* and *D. mauritiana* [17,43].

Fourth, we tested a single concentration for the two acids (4.6 g·L^−1^) which discriminates oviposition site choice behavior between the four species of the melanogaster complex including *D. sechellia* and *D. melanogaster* [17]. We confirmed that these two species show different feeding behavior at this concentration but generalist and specialist *D. yakuba* did not seem to discriminate them. However, at a higher concentration of hexanoic acid (HA), a significant difference between generalist and specialist *D. yakuba* was found. In *D. melanogaster*, the concentration-dependent gustatory response to hexanoic acid is complex and depends on interactions between sweet- and bitter-tasting neurons [22]. Whereas, at concentration < 9.3 g·L^−1^ sweet-tasting neurons elicit an appetitive behavior, at higher concentrations bitter-tasting neurons induce a repulsive behavior. Given that most of the selection signal we identified here for *D. yakuba* was on bitter-tasting gustatory receptor genes, our behavioral assays suggest that those genes might have played a role in a reduced acid-repulsion in *D. y. mayottensis*.

Despite these experimental considerations, we were still able to recover a positive correlation between the results from the olfactory experiment and assays including the two carboxylic acids characteristic of the ripe stage of noni.

### 4.2. Apparent Lack of Selection on Acid-Sensing Ionotropic Receptor Genes

Octanoic and hexanoic acids constitute ~60% and 20% of the total volatiles of ripe noni fruits, respectively [15]. As rotting proceeds, these acids degrade and the concentration of their respective octanoate and hexanoate esters (e.g., methyl hexanoate) increases [44]. Generalist *Drosophila* are usually attracted to acetic acid produced by bacteria developed on rotten, fermented fruits, as well as by its derived acetate esters produced by the fermenting yeasts [45]. Specialization on noni in *D. sechellia* and *D. y. mayottensis* should have involved increasing responsiveness to medium-chained carboxylic acids and their ester derivatives in couple with decreasing responsiveness to short-chained carboxylic acids and their ester derivatives.

Prieto-Godino et al. [19] have recently shown that increased responsiveness to hexanoic (but not octanoic) acid in *D. sechellia* has involved coding and *cis*- and *trans*-regulatory changes in the ionotropic receptor *Ir75b*, which is responsive to butyric acid in its generalist relatives. They have also suggested that *cis*-regulatory changes expanding *Ir75b* have evolved in the ancestor of *D. sechellia* and *D. simulans* but have probably remained silent until trans-regulatory and coding changes occurred in the *D. sechellia* lineage. The increase of *Ir75b* responsiveness was accompanied by a decrease in responsiveness of the tandem paralog *Ir75a* to acetic acid [46]. Remarkably, we did not detect any signal of selection on these two genes in *D. yakuba*. Moreover, we found that, unlike *D. melanogaster*, both generalist and specialist *D. yakuba* did not significantly differ from *D. sechellia* in hexanoic acid preference, further suggesting that this acid-sensing pathway might not have played an important role in *D. yakuba* specialization.

### 4.3. Variation in the Ester-Sensing Olfactory Receptor Or22a in Populations from the Ancestral Range

Dekker et al. [18] showed that *D. sechellia* antennae responded most strongly to noni esters (methyl hexanoate) than to the two acids. They found that *D. sechellia* sensitivity to methyl hexanoate was also present in its closest relative *D. simulans* and correlated with an expansion of ab3 sensilla that are more sensitive to ethyl hexanoate in *D. melanogaster* and *D. yakuba*. However, both coding sequence and transcription level comparisons of *Or22a*, the main receptor of ab3 sensilla, suggest that hypersensitivity to methyl hexanoate might have evolved in the ancestor of *D. sechellia* and *D. simulans* [18,46,47]. Using elegant transgenic experiments in *D. sechellia*, Auer et al. [20] have recently confirmed the role of a coding change in *Or22a* in the olfactory response to noni fruits and provided evidence for the ancestry of that change in the simulans species complex.

Remarkably, we found strong allelic differentiation between *D. y. mayottensis* and *D. y. yakuba* from Kenya at *Or22a*, mostly including coding changes. However, closer examination of sequence evolution of this gene in *Or22a*, indicated that variants in both populations are also found at intermediate frequencies in the Cameroon population of *D. y. yakuba*, which likely represents the ancestral range of the species [40]. Indeed, Auer et al. [20] examined four geographical lines of *D. yakuba* including the three populations studied here as well as the reference genome line from Côte d’Ivoire. They found that the *D. y. mayottensis* line did not show significant differences in number of Or22a-expressing olfactory sensory neurons and their physiological responses to ethyl and methyl esters in spite of differences in their protein sequences. Given the level of polymorphism at this gene within and between *D. yakuba* populations that we found here, additional lines from Cameroon and other populations from West Africa should be tested in the future to fully understand the potential role of ancestral *Or22a* variation in driving the specialization of *D. y. mayottensis* on noni.

### 4.4. Selection on Bitter-Sensing Gustatory Receptors

Unlike olfactory perception, the role of gustatory receptor evolution in noni specialization remains unclear. Indeed, previous comparative genomics studies have revealed faster protein evolution and turnover for gustatory receptor genes than for olfactory receptors [6,7], but a functional link remains to be identified. Our phenotypic assay found no significant difference in feeding preference between *D. melanogaster* and *D. sechellia* for the methyl hexanoate ester. This may suggest that acids, not esters, are the most important constituents for the gustatory specialization on noni. In *D. melanogaster*, appetitive response to low concentration hexanoic and octanoic acids is induced by the gustatory ionotropic receptors *Ir25a*, *Ir76b* and *Ir56d* in sweet-tasting sensilla [22]. However, at high concentrations of hexanoic acids (>9.3 g·L^−1^), this behavior is suppressed by the activation of bitter-sensing sensilla independent of the three ionotropic receptors [22]. We did not detect any deviation from neutral evolution on these ionotropic receptor genes. Instead, we detected a signal of selection on four bitter-tasting gustatory receptors *Gr22b*, *Gr22d*, *Gr59a* and *Gr93c*. Because generalist *D. y. yakuba* significantly differs in its preference for hexanoic acid (at 1%) from the specialist *D. sechellia*, whereas specialist *D. y. mayottensis* does not, the four gustatory receptors might have evolved to reduce *D. y. mayottensis* avoidance of high-concentration hexanoic acid. Unlike olfactory receptors whose evolution is strongly modular, gustatory receptors act in an integrative way so that changes in one receptor unpredictably change the response of an entire set of receptors to the same component [39]. For example, and most relevant to our findings, Sung et al. [47] showed epistatic interactions between loss-of-function mutations of the gustatory receptors *Gr22e* and *Gr59c*, two paralogs of our detected receptors, that depended on the tested bitter substance. In *D. sechellia*, three out of the five genes of the Gr22a clade were pseudogenized, namely *Gr22f*, *Gr22c* and *Gr22d* [6]. Although none of the most differentiating sites at the Gr22a clade in *D. yakuba* affects the protein sequence, we found a nonsense mutation at *Gr22d* that was nearly fixed in *D. y. mayottensis* whereas segregating at low frequencies in *D. y. yakuba*. Because nonsense mutations could be post-transcriptionnally corrected in neural cells [46], the role of this mutation remains to be tested. Future functional genetic analyses of the genes identified here would definitively shed light on their potential effects. Those future gustatory analyses should also include either whole noni fruits, or other noni chemical components that could be perceived as bitter by *D. y. mayottensis*. Nonetheless, the significant difference found in *D. melanogaster* iso-1 strain, which carries nonsense mutations at both *Gr22b* and *Gr22d*, to hexanoic acid indeed suggests the involvement of those genes in noni toxins perception.

In summary, our results support a role for standing genetic variation in promoting host plant preference evolution in herbivorous insects. Because such variation may be abundant in several insect-plant systems [48], its probable maintenance by various forms of balancing selection in ancestral populations may be an initial prerequisite for rapid host shifts, facilitating the gradual built-up of adaptations in simultaneous host use traits. Future analyses should therefore focus on how standing variation in preference alleles accumulate and interact with host performance traits at the onset of ecological specialization and adaptation.

## Figures and Tables

**Figure 1 genes-12-00032-f001:**
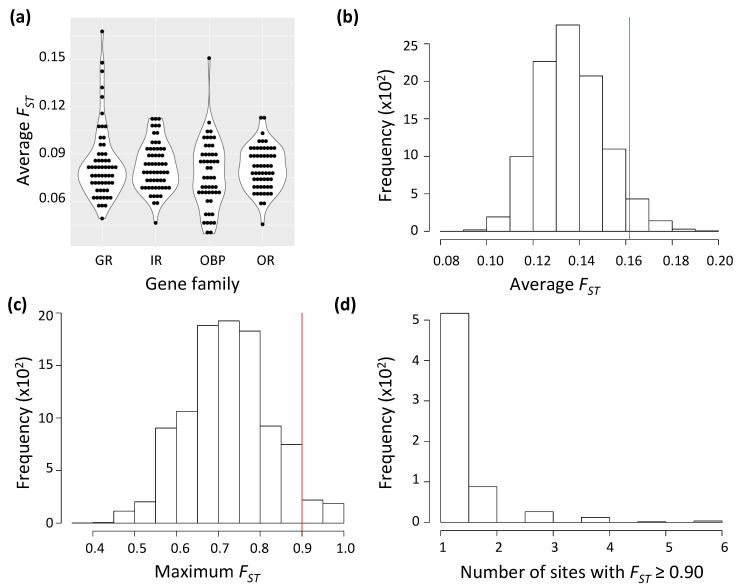
(**a**) Kernel density distributions of genetic differentiation (*F_ST_*) estimates at chemosensory genes (±2 kb) between specialist *Drosophila yakuba mayottensis* from Mayotte and generalist *D. y. yakuba* from Kenya. Gene families are abbreviated as: GR = gustatory receptors, IR = ionotropic receptors, OBP = odorant-binding-proteins and OR = olfactory receptors. (**b–d**) Histograms of the outcome of the *msms* simulations based on DADI-inferred demographic model based on presumably neutral autosomal short intronic sequences for (**b**) average *F_ST_*, (**c**) maximum *F_ST_*, and (**d**) number of sites with *F_ST_* ≥ 0.9 per run. Vertical red lines indicate the 0.95 quantile *F_ST_* value.

**Figure 2 genes-12-00032-f002:**
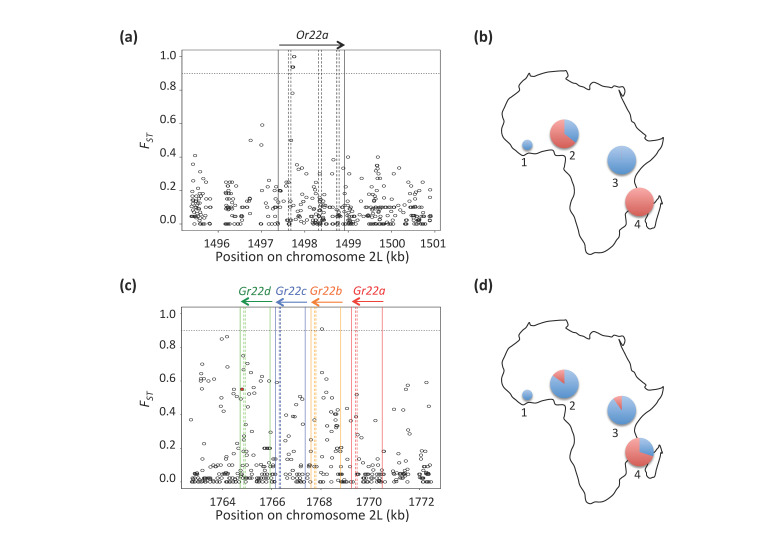
Genetic differentiation (*F_ST_*) profile between *D. y. mayottensis* and *D. y. yakuba* (Kenya) at (**a**) the olfactory receptor *Or22a* gene, with (**b**) the geographical distribution of its most differentiating site in Mayotte, Kenya, Cameroon and the reference genome strain from Côte d’Ivoire, and at (**c**) the gustatory receptor genes of the Gr22a clade showing the non-sense mutation at *Gr22d* (red dot) and (**d**) its geographical distribution. Solid and dashed vertical lines indicate gene and exon boundaries, and arrows indicate the gene sense. Dotted horizontal line indicates the *F_ST_* value corresponding to the 0.95 quantile of the *msms* simulations.

**Figure 3 genes-12-00032-f003:**
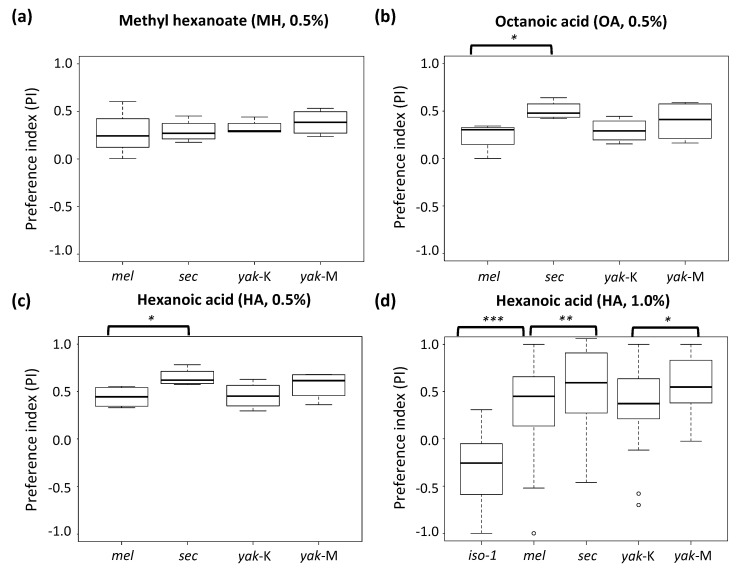
Preference indices of the feeding behavior assays for (**a**) methyl hexanoate at 0.5%, (**b**) octanoic acid at 0.5%, and (**c**) hexanoic acid at 0.5% and (**d**) 1.0%. Strain names are abbreviated as: *mel* = *D. melanogaster* from Mt Oku, iso-1 = *D. melanogaster* strain mutant for *Gr22b* and *Gr22d*, *sec* = *D. sechellia*, *yak*-K = *D. yakuba yakuba* from Kunden and *yak*-M from *D. y. mayottensis*. *p*-values are indicated as: * < 0.05, ** < 0.01 and *** < 0.001.

**Table 1 genes-12-00032-t001:** List of most differentiating sites in chemosensory genes with *F_ST_* values ≥ 0.86 between Kenya and Mayotte populations, corresponding to the 0.90 quantile value estimated from the *msms* simulation.

Position	Alleles (ref./alt.)	Gene	Effect	Allele Frequencies			*F_ST_*
				Cameroon	Kenya	Mayotte	
X:10060167	G/A	Or13a	Downstream	8/4	1/11	22/0	0.917
X:13654476	T/A	Ir7a	Downstream	5/9	10/0	4/40	0.909
X:13654477	C/A	Ir7a	Downstream	5/9	10/0	4/40	0.909
2L:1497713	G/T	Or22a	Synonymous	6/10	15/1	0/44	0.938
2L:1497721	C/A	Or22a	Synonymous	6/10	15/1	0/44	0.938
2L:1497723	T/G	Or22a	Nonsynonymous (V/G)	6/10	15/1	0/44	0.938
2L:1497724	A/G	Or22a	Nonsynonymous (V/G)	6/10	15/1	0/44	0.938
2L:1497730	C/T	Or22a	Synonymous	6/14	15/1	0/44	0.938
2L:1497751	A/G	Or22a	Nonsynonymous (I/M)	9/5	16/0	0/44	1.000
2L:1497754	T/C	Or22a	Synonymous	9/5	16/0	0/44	1.000
2L:1497757	T/C	Or22a	Synonymous	5/13	16/0	0/44	1.000
2L:1764188	A/C	Gr22d	Downstream	4/14	0/20	19/3	0.864
2L:1768039	T/A	Gr22b	Synonymous	20/0	20/0	2/20	0.909
2R:18929879	G/A	Gr59a	Upstream	16/4	20/0	5/39	0.886
3R:18490921	G/A	Gr93b	Downstream	16/2	20/0	3/19	0.864
3R:18494463	A/G	Gr93c	Nonsynonymous (F/S)	12/8	18/0	4/40	0.909

## Data Availability

Experimental data could be obtained from the authors upon request.

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
