# Peer review of "Soft Selective Sweep on Chemosensory Genes Correlates with Ancestral Preference for Toxic Noni in a Specialist Drosophila Population"

_genes, 2020, doi:10.3390/genes12010032_

Round 1

Reviewer 1 Report

This interesting study suggests the input of ancestral genetic variation in host plant selection by Drosophila species. It builds on previous analysis by the same group.

I cannot comment on the design and appropriateness of the in silico methodologies, and instead focus on the results and conclusions.

major changes:

line 177-178 define the species this sentence refers to

line 184-185 'It was only followed by its two adjacent paralogs Gr22b (FST = 0.143) and Gr22d (FST = 0.147).' I do not understand what the word 'only' implies.

All Graphs: the labelling of axes was very unclear in the pdf version under review.

line 243-244 & line 385-387 a nonsense mutation is NOT a missense mutation. Also the phrase 'the most differentiating site falls ~0.5 kb downstream the gene' is misleading when describing a site within the gene (ie in the 2nd exon). In addition, having highlighted Gr22c in 3.1, the nature of polymorphisms in this gene is not described. Is there any evidence for translation read-through of nonsense codons in neural cells?

Minor changes:

line 109 'a list'

line 110 'to the D. melanogaster genome'

line 111 'to choose'

line 240 'motivated us to...'

Author Response

Reviewer 1

This interesting study suggests the input of ancestral genetic variation in host plant selection by Drosophila species. It builds on previous analysis by the same group.

I cannot comment on the design and appropriateness of the in silico methodologies, and instead focus on the results and conclusions.

major changes:

line 177-178 define the species this sentence refers to

Species name “D. yakuba” is added.

line 184-185 'It was only followed by its two adjacent paralogs Gr22b (FST = 0.143) and Gr22d (FST = 0.147).' I do not understand what the word 'only' implies.

“only” is removed.

All Graphs: the labelling of axes was very unclear in the pdf version under review.

Following the journal’s submission format, figures had to be embedded within a .doc template file. This might have affected their resolution during conversion to pdf. We suppose that higher resolution figures will be independently submitted if the manuscript is accepted on the journal’s site.

line 243-244 & line 385-387 a nonsense mutation is NOT a missense mutation.

We thank the reviewer for this correction and correct the text.

Also the phrase 'the most differentiating site falls ~0.5 kb downstream the gene' is misleading when describing a site within the gene (ie in the 2nd exon).

The most differentiating site indeed falls downstream Gr22d as shown in Figure 2c, whereas Gr22d nonsense mutation (red dot in Figure 2c) falls within the second exon. We rephrase the sentence by replacing “adjacent” by “close”.

In addition, having highlighted Gr22c in 3.1, the nature of polymorphisms in this gene is not described.

Because our gene-level analyses also included 2kb up- and downstream sequences for each gene, the high average FST values for Gr22c that we found above is mostly due to the high differentiation sites in Gr22b and Gr22d, since the transcribed sequence of this gene does not include any particularly differentiating sites. We add this explanation to the paragraph.

Is there any evidence for translation read-through of nonsense codons in neural cells?

As stated in the Discussion, the only evidence of translation read-through of nonsense codons in neural cells comes from the study of Ir75a receptors in D. sechellia (Prieto-Godino et al. 2016 Nature). Whether this also occurs in gustatory receptors is, to the best of our knowledge, not known. We replaced “can” by “could”.

Minor changes:

line 109 'a list'

Corrected.

line 110 'to the D. melanogaster genome'

Corrected.

line 111 'to choose'

Corrected.

line 240 'motivated us to...'

Corrected.

Reviewer 2 Report

The authors investigated a possible soft selective sweep theory of gustatory receptor genes in two Drosophila species.

The manuscript is well written and represents a step forward to understand the adaptative selection of gustatory genes in Drosophila populations. However, I have some concerns about how the evidence is presented in the manuscript.

  1. Lack of enough background in the introduction. The authors need to give to the readers enough evidence of previous studies on gustatory receptor genes without the need to read previous publications from the authors. I suggest that introducing what gustatory receptor genes affect population differentiation, genes associated with them and why D. yakuba and D. sechellia are important is necessary to understand why the authors decided to study these genes.
  2. No connection between behavior experiments and soft selective sweep analysis. Why the authors did not focus on bitter-tasting behavior? Looks like that both analyses are not connected. Please give evidence on how acid-tasting behavior could be somehow related to bitter-tasting behavior.
  3. The discussion focused mainly on acid-tasting behavior rather than bitter-tasting genes and behavior. I suggest the authors focus on their findings.

Minor comments:

the perl script should be publicly available.

Author Response

The authors investigated a possible soft selective sweep theory of gustatory receptor genes in two Drosophila species.

The manuscript is well written and represents a step forward to understand the adaptative selection of gustatory genes in Drosophila populations. However, I have some concerns about how the evidence is presented in the manuscript.

Lack of enough background in the introduction. The authors need to give to the readers enough evidence of previous studies on gustatory receptor genes without the need to read previous publications from the authors. I suggest that introducing what gustatory receptor genes affect population differentiation, genes associated with them and why D. yakuba and D. sechellia are important is necessary to understand why the authors decided to study these genes.

The Introduction has been revised in agreement with Reviewer 2’s recommendations. A paragraph detailing the study background was added. It reads:

“Of particular interest is the parallel specialization of the species D. sechellia and the subspecies D. yakuba mayottensis on toxic noni (Morinda citrifolia) fruits on separate islands of the Indian Ocean, i.e. the Seychelles and Mayotte, respectively [13,14]. Noni toxicity is due to its high content of medium-chained carboxylic acids (hexanoic and octanoic acids) in its fruit [15]. Remarkably, D. sechellia has evolved a strong olfactory attraction to those acids [16,17], as well as to methyl hexanoate, the major ester characteristic of the rotten, non-toxic fruit of noni [18]. Genetic studies revealed that evolutionary changes at the ionotropic receptor Ir75b and at the olfactory receptor Or22a underlie D. secnehllia preference to noni hexanoic acid and methyl hexanoate, respectively [19,20]. Matsuo et al. [21] also suggested that the odorant-binding-protein OBP57e/d may also play a role in the gustatory preference of this species to noni acids. However, in spite of evidence for rapid evolution of gustatory receptors in D. sechellia[6,7], the role of those receptors in noni specialization remains unknown. This may be partly because both sugar and bitter sensing neurons may be involved in dose-dependent acid sensing, as it has recently been demonstrated from functional studies in D. melanogaster [22], therefore complicating the dissection of this character. For D. y. mayottensis, an olfactory preference for noni fruits in adult flies exists [14], but we still do not understand of the chemosensory evolution underlying this preference, as well as possible gustatory preference, in this subspecies.”

No connection between behavior experiments and soft selective sweep analysis. Why the authors did not focus on bitter-tasting behavior? Looks like that both analyses are not connected. Please give evidence on how acid-tasting behavior could be somehow related to bitter-tasting behavior.

The discussion focused mainly on acid-tasting behavior rather than bitter-tasting genes and behavior. I suggest the authors focus on their findings.

As previously stated in the Discussion and now added to the Introduction, acid-sensing in D. melanogaster depends on both sweet- and bitter-sensing gustatory receptor neurons (Ahn et al. 2017 eLife). We rephrased a paragraph in the Discussion to make the link between acid- and bitter-tasting, and consequently between findings from the population genomics analysis and the behavioral assays clearer as follows:

“In D. melanogaster, the concentration-dependent gustatory response to hexanoic acid is complex and depends on interactions between sweet- and bitter-tasting neurons [22]. Whereas, at concentration < 9.3 g.L-1 sweet-tasting neurons elicit an appetitive behavior, at higher concentrations bitter-tasting neurons induce a repulsive behavior. Given that most of the selection signal we identified here for D. yakuba was on bitter-tasting gustatory receptor genes, our behavioral assays suggest that those genes might have played a role in a reduced acid-repulsion in D. y. mayottensis.”

However, we agree with Reviewer 2 that other bitter components in noni fruits may also be perceived by the rapidly-evolving receptors. We therefore added to the Discussion:

Those future gustatory analyses should also include either whole noni fruits, or other noni chemical components that could be perceived as bitter by D. y. mayottensis. Nonetheless, the significant difference found in D. melanogaster iso-1 strain, which carries nonsense mutations at both Gr22b and Gr22d, to hexanoic acid indeed suggests the involvement of those genes in noni toxins perception.”

Minor comments:

the perl script should be publicly available.

The scripts were added as a supplementary text.

Round 2

Reviewer 2 Report

Thank you to the authors for addressing my concerns. I recommend the manuscript for publication.